# Synbiotic-Glyconutrient Additive Reveals a Conducive Effect on Growth Performance, Fatty Acid Profile, Sensory Characteristics, and Texture Profile Analysis in Finishing Pig

**DOI:** 10.3390/foods13010105

**Published:** 2023-12-28

**Authors:** Vetriselvi Sampath, Jae Hong Park, In Ho Kim

**Affiliations:** 1Department of Animal Resource and Science, Dankook University, Cheonan 330-714, Chungnam, Republic of Korea; suve2314@gmail.com (V.S.); atom1965@hanmail.net (J.H.P.); 2Smart Animal Bio Institute, Dankook University, Cheonan 330-714, Chungnam, Republic of Korea

**Keywords:** growth performance, fatty acid profile, sensory characteristics, synbiotic-glyconutrient

## Abstract

This study aims to investigate the effect of synbiotic-glyconutrients (SB-GLN) additive on growth performance, fatty acid profile, sensory characteristics, and texture profile analysis in finishing pig. Landrace × Yorkshire ♀ × (Duroc ♂) (*n* = 60) pigs with average body weight of 54.88 ± 1 kg were allocated into one of three dietary treatment groups in a complete randomized block design with four replicates of five pigs (two barrows and three gilts) per pen. The test treatments (TRT) were CON—corn-soybean meal basal diet; TRT 1—CON+ 0.25% SB-GLN; and TRT 2—CON + 0.5% SB-GLN. SB-GLN contains 1 × 10^7^ CFU/g each of: *L. plantarum*, *B. subtilis*, and *S. Cerevisiae*, and 5% yeast cell wall β-Glucans (from S. Cerevisiae), and 14% of glyconutrients (N-acetylglucosamine, D-xylose, and Fucose). Pigs fed SB-GLN supplement showed linearly increased (*p* < 0.05) body weight, daily gain, and daily feed at the end of week 5, 10, and the overall experimental period. In addition, G:F showed a tendency to decrease (*p* < 0.1) at the end of week 10 and the overall experimental period. In addition, pigs that received a graded level of SB-GLN showed a tendency to increase (*p* < 0.1) their longiness muscle area and decreased (*p* < 0.05) cooking loss. The sensory results of pork belly (tenderness and juiciness) and loin (flavor) meat, and the texture profile analysis parameters of hardness 1, cohesiveness, and gumminess (belly), and hardness 2, chewiness, and springiness (loin) meat were linearly higher (*p* < 0.05) in the SB-GLN group. The values of fatty acid like butyric acid, caproic acid, undecylic acid, tridecylic acid, myristic acid, pentadecyclic acid, palmitic acid, margaric acid, stearic acid, eicosapentaenoic acid, and lignoceric acid were higher in pork belly fat of the SB-GLN-treated group compared to CON. Moreover, pigs that received SB-GLN exhibited higher crude fat and lauric acid, myristic acid, pentacyclic acid, palmitic acid, margaric acid, Octadecanoic acid, Oleic acid, linoleic acid, and eicosapentaenoic acid FA profiles in belly-lean meat. Also, the FA profile of the SB-GLN-treated group loin-lean meat showed increased lauric acid, myristic acid, palmitic acid, margaric acid, stearic acid, oleic acid, linoleic acid, alpha-linoleic acid, and eicosapentaenoic acid. The SB-GLN-treated group pork belly fat, belly lean meat, and loin-lean meat showed linearly increased docosahexaenoic acid, nervonic acid, omega 3, omega 6, ω-6: ω-3, Σ saturated FA, Σ un-SFA, Σ mono-USFA, Σ poly-USFA, MUFA/SFA, and PUFA/SFA. Therefore, we infer that the inclusion of 0.5% SB-GLN additive to finishing pig diet would be more beneficial to enhance their performance, and to increase the essential FA profile of pork meat for human consumption.

## 1. Introduction

Pork meat has gained significant prominence and emerged as a highly consumed meat in Asia and Europe [1] due to its high protein content and essential fatty acids, particularly saturated fatty acids, which are intricately linked to human health considerations [2]. Recently, there has been increasing interest in how to produce high-quality pork. Indeed, the quality of pork can be evaluated by various characteristics such as sensory traits, intramuscular fat (IMF) content, and fatty acid composition. IMF is primarily distributed in the layers of skeletal muscle (epimysium, perimysium, and endomysium) and has a positive correlation with meat quality such as flavor, tenderness, and juiciness [3]. Diets, genetics, environment, management practices, and production systems [4] can influence the meat quality and thus it is highly ideal to explore effective strategies to enhance the pork quality. Dietary intervention is one of the most common methods to improve the performance of animal and to increase the meat quality.

Prebiotics (yeast cell walls and mannan polysaccharides) are defined as “a nondigestible food ingredients [5] that beneficially affects the host by selectively stimulating the growth and/or activity of one or a limited number of bacteria in the colon” [6]. Probiotics, a “live microbial feed additive”, are known to improve the performance of the host by improving their gut microbial balance [7]. Some studies indicate that probiotics could modify the muscle fatty acid (FA) profile in broilers [8,9]. The combination of pre-and probiotic in the form of synergism is known as “synbiotics” [10]. Such a synergistic synbiotic contains a substrate that is selectively utilized by co-administered microorganism(s) [11]. The primary reason for using synbiotics in food products is that probiotics do not survive well in the digestive tract without prebiotic foods. Without the necessary food source for probiotic bacteria, their tolerance to temperature, oxygen, and pH level may decrease [12]. Previously, Hassanpour et al. [13] demonstrated that broilers fed a diet supplemented with synbiotics had better feed efficiency. Similarly, Ghasemi et al. [14] pointed that the inclusion of 1 g/kg synbiotics to the broiler diet significantly decreased plasma cholesterol and LDL concentrations. Glyconutrients (plant sugars) are rich in anti-inflammatory and antimicrobials properties. They can increase the energy efficiency and health of the host and promote their cellular integrity [15]. Earlier studies have demonstrated the beneficial effects of pre and probiotics in the monogastric animal diet. For instance, De Vries et al. [16] noted that inclusion of yeast cell wall ß-Glucans significantly increased the gut health of pigs. Similarly, Awad et al. [17] reported that dietary inclusion of synbiotics at a concentration of 1 g/kg improved the body weight gain and feed efficacy in broilers. On the other hand, Lee et al. [18] noted that dietary inclusion of synbiotics containing a probiotic originating from anaerobic microbiota (bacteria—10^9^ CFU/mL, yeast—10^5^ CFU/mL, molds—10^3^ CFU/mL) and a prebiotic (MOS, sodium acetate, ammonia citrate) has improved digestion of nutrients in weaning pigs. Additionally, Aksu et al. [19] stated that broilers fed a diet supplemented with probiotics showed increased lipid oxidation stability, water-holding capacity, tenderness, and sensory properties, while Zhang et al. [20] found no impact on chicken meat with pre-and probiotic supplements. So far, several studies addressed the combination effect of synbiotics and glyconutrients (SB-GLN) in ruminants’ performance and meat quality. For example, Valencia et al. [21] reported that a combination of probiotic-glyconutrient has increased live weight gain, decreased the mortality rate, and lowered the non-esterified fatty acids in Holstein calves. Similarly, Castro-Perez et al. [22] noted that dietary SB-GLN improved the growth performance and carcass weight in lambs. The above-mentioned literature has provoked us to hypothesize and initiate this research to know whether the inclusion of the SB-GLN combination could enhance the growth performance, fatty acid profile, and meat quality of pigs or not. Therefore, the goal of this research was to examine the growth performance, fatty acid profile, and the quality of pork meat by adding an SB-GLN additive to finishing pigs’ diet.

## 2. Materials and Methods

### 2.1. Ethics

The present study was carried out at Dankook University “Swine research center” (Gongju, Republic of Korea)”. The husbandry practices adhered to animal welfare guidelines, and the research protocol (No: DK-2-2221) received approval from the Institutional Animal Care and Use Committee (IACUC) of Dankook University (Cheonan, Republic of Korea) before the commencement of the trial.

### 2.2. Experimental Design, Animals, and Management

A total of 60 crossbred [(Landrace × Yorkshire ♀  ×  (Duroc ♂] finishing pigs were used in this 10-week trial. The initial average body weight (IBW) of all pigs was adjusted to approximately 54.88 ± 1 kg. There were three treatment groups. Each treatment had 4 replicates and 5 pigs (2 barrows and 3 gilts) per pen. The finishing facility was equipped with natural ventilation, slatted concrete floors, and the barn temperature was fixed at 21.5 °C. Feeders and water dispensers were set in the corners of each pen measuring 1.8 m × 1.8 m. This arrangement provided pigs unrestricted access to both feed and water throughout the trial. The breeding room was monitored by the trainees three times a day (9:00 A.M., 2:00 P.M., and 7:00 P.M.) to check any leakage in the water trough, sufficient feed in the feeder, and occurrence of health issues.

### 2.3. Experimental Diets and Dietary Regimen

Based on IBW (54.88 ± 1 kg) and gender (barrow and gilt), pigs were randomly allocated into one of three dietary treatment groups: CON—Corn-soybean meal basal diet; TRT 1—CON+ 0.25% SB-GLN; and TRT 2—CON + 0.5% SB-GLN in a complete randomized block design. The basal diet (mash form) was formulated to according to NRC regulation [23] (Table 1), while SB-GLN was commercially procured from Nongh-yup Feed Inc., Seoul (South Korea); it contains 1 × 10^7^ CFU/g each of: *L. plantarum*, *B. subtilis*, and *S. Cerevisiae*, and 5% yeast cell wall ß-Glucans (from *S. Cerevisiae*), and 14% of glyconutrients (N-acetylglucosamine, D-xylose, and Fucose). The test ingredients were mixed using a DDK-801(Daedong Tech, Siheung, Republic of Korea) mixer and stored in pre-marked bags.

### 2.4. Sampling and Analysis

#### 2.4.1. Growth Performance

The growth performance traits such as average daily gain (ADG), daily feed intake (ADFI), and gain to feed ratio (G:F) were recorded at the end of weeks 5, 10, and the overall experimental period. Pigs’ body weight (BW) was measured individually at the start of the experiment and at the end of weeks 5 and 10 using a G-Tech GL-6000L portable bench scale (Republic of Korea) to determine their ADG. The feeder was filled at 9:00 A.M. and the scrapings in the feeders were collected and weighed at 5:00 P.M. to calculate the ADFI. G:F was determined by dividing ADFI and ADG.

#### 2.4.2. Meat Quality

At the end of week 10, 24 pigs (2 pigs/pen) were transported to a local slaughterhouse and rested for 6 h. During lairage time, animals were fasted with *ad libitum* access to water. CO_2_ stunning was performed prior to slaughter. After slaughter, chilled (20 °C ± 1 °C) carcasses were transported to a cutting room. The longissimus thoracis et lumborum (LTL) muscles were collected from the left carcasses side, and all visual fats and connective tissues were trimmed off and made into sub-samples for further analyses. The LTL muscle area, meat color, pH, water-holding capacity (WHC), drip loss, and cooking loss were measured 24 h post-mortem (Figure 1). Remaining samples were stored at –20 °C for fatty acid profile, texture profile, and sensory evaluation. The LTL muscle area was measured by the digitization area-line sensor. The meat color, marbling, and firmness scores were evaluated according to National Pork Producers Council [24]. Prior to checking the meat color, the Konica Minolta CR-400 Chroma meter (Konica Minolta Sensing Americas Inc., Ramsey, NJ, USA) was standardized with a white plate (Y = 86.3, X = 0.3165, and y = 0.3242). Sample color was expressed according to the Commission International de l’Eclairage (CIE) system and described as L* (lightness), a* (redness), b* (yellowness), Chroma, and hue angle (h°). The chroma angle was calculated as (a*^2^ + b*^2^)^0.5^, while hue angle was calculated as tan^–1^ (b*/a*). The pH value of the sample (both side) was measured by the digital pH probe (NWK-Technology GmbH, Kaufering, Germany). For water-holding capacity (WHC) analysis, 0.3 g of the sample was placed in the middle of 120 mm filter paper and pressed for 3 min. Areas of the compressed sample and the expressed humidity were defined and determined by using a digitalized area-line sensor (MT-10S, M.T. Precision Co., Ltd., Tokyo, Japan). The ratio of water: meat area was then calculated, giving a measure of WHC (a smaller ratio indicates increased WHC) [25]. Drip loss and cooking loss was determined following the methods of Choe et al. [26].

#### 2.4.3. Texture Profile Analysis

*Latissimus dorsi* (belly) and LTL muscle (loins) samples (Figure 2) were defrosted overnight at 4 °C, and cut into 50 mm thick chops (*n* = 8/trt) without fat or connective tissue parallel to the longitudinal orientation of the muscle fibers. Then the fiber axis of the sample was perpendicular to the direction of the probe. Texture Profile analysis (TPA) was performed with raw meat using a TAXT2i texture analyzer (Stable Micro System, Godalming, UK). In brief: meat specimens were placed under a cylindrical probe and moved downwards at a constant speed of 3.0 mm/s (pre-test), 1.0 mm/s (test) and 3.0 mm/s (post-test). The probe constantly moved downwards until piercing a predetermined percentage of the sample thickness, retracted to the initial point of contact with the sample, and stopped 2 s before initiation of the second compression cycle. During the test, the force of the sample was recorded every 0.01 s and plotted on a force time plot [27]. Finally, TPA parameters (hardness, cohesiveness, adhesiveness, gumminess, fracture, stringiness, chewiness, and springiness index) were calculated following the standard procedure of Honikel [26].

#### 2.4.4. Sensory Evaluation

The sensory evaluation (SE) of pork meat was conducted in individual stalls with white lighting. Four panels were randomly allocated for SE test. These panels (*n* = 12, ♀ and ♂) were trained with finely tuned sensory perceptions. At first, the vacuum-packed samples were thawed for 2 h at 4 °C. Then the representative samples were sliced to 15 mm. For SE color test, freshly cut slices (30 min) from each sample were passed through to all panelists. The remaining samples were cooked by grilling (TECHEF Stovetop Korean BBQ Non-Stick Grill Pan) without salt or spices for 2 min and flipped 30 s intervals. The temperature during cooking was consistently kept at approximately 220 °C as observed and maintained with the use of an infrared thermometer. Immediately after cooking, the samples were transferred to the serving boat with a three-digit random code and distributed to the panelists. A 5 min interval was given to the panels between the evaluations of each sample and they were instructed to cleanse their palate with distilled water and have salt-free crackers. Finally, SE traits like tenderness, flavor, juicy, texture, and preference (overall acceptability) were evaluated using a 7-point scale as described by Ba et al. [28].

#### 2.4.5. Fatty Acid Analyses

The fat content in the samples was extracted using chloroform: methanol (2:1) solvent mixture (SM). In brief: the samples were grinded at first, weighed (15 g), and mixed with 150 mL of SE at 300× *g* for 3 min using PT-MRC 2100 (Littau, Switzerland) homogenizer. Later, samples were filtered using Whatman filter paper. Then, 20 g of Sodium Sulfate solution was added to the filtered solution and mixed for 1 min and the upper fat layer was transferred to a titration flask. Then the samples were allowed to dry at 55 °C using a rotary evaporator and the fat layer was reconstituted with 1 mL tricosylic acid and 1 mL of 0.5 N sodium hydroxide and converted to fatty acid methyl esters (FAME). About 1:0 mL of the FAME sample was taken and kept in an auto-sampler vial, sealed, and used for FA analysis. The FAMEs separation was successfully accomplished by utilizing a gas chromatography/flame ionization detector (GC-FID, Columbia, MD, USA). This GC FID system was equipped with an Omega wax capillary column, which measured 30 m in length, 0.25 mm in diameter, and had a film thickness of 0.25 μm. The oven temperature was maintained at 50 °C for 1 min, and ramped at a rate of 25 °C/min to 200 °C, and further raised at a rate of 5 °C/min to 230 °C. The injection and detector temperatures were set to 250 ° C. Finally, FA in the samples was determined by comparing their retention times to those acquired from standard FA. Each individual FA was then quantified and expressed as a percentage relative to the total FA present in the samples. The Omega 3 to omega 6 PUFA ratio (*n*-6/*n*-3) was estimated. The following FA ratios: palmitoleic isomers to palmitic acid (C16:1/C16:0), oleic to stearic acids (C18:1/C18:0), dihomo-γ-linolenic to linoleic acid (C20:3 *n*-6/C18:2 *n*-6), docosapentaenoic to adrenic acid (C22:5 *n*-3/C22:6 *n*-3), and arachidonic to linoleic acid (C20:4 *n*-6/C18:2 *n*-6) were estimated according to Boschetti et al. [29].

#### 2.4.6. Statistical Analysis

Experimental data were investigated using General Linear Model (GLM) procedure of SAS (SAS, Inst. Inc., Cary, NC, USA) in a complete randomized block design. Performance variables were analyzed with the pen as an experimental unit, while meat quality, FA, TPA, and SE were analyzed using the individual pig as an experimental unit. Orthogonal polynomial comparisons were conducted to determine linear and quadratic effects of 0%, 0.25%, and 0.5% SB-GYL supplement in pigs’ diet. Differences among treatment means were determined using Tukey’s range test and the probability value of less than 0.05 and 0.10 was considered significant and trend, respectively.

## 3. Results

The effect of SB-GLN on growth performance of finishing pigs is shown in Table 2. Pigs fed a diet supplemented with SB-GLN showed a linear increase in (*p* < 0.05) BW at the end of weeks 5 and 10. Compared to the CON group, the SB-GLN-treated group pigs showed higher (*p* < 0.05) ADG and ADFI at the end of weeks 5 and 10. Moreover, dietary supplement with SB-GLN showed linearly increased (*p* < 0.05) ADG and ADFI, and tended to decrease (*p* < 0.1) the G:F ratio during the overall experimental period.

The effect of SB-GLN on finishing pig meat quality is shown in Table 3. Pigs that received a graded level of SB-GLN showed a tendency to increase (*p* < 0.1) LTL muscle area, and linearly decreased (*p* < 0.05) cooking loss compared to CON. Whereas at the end of week 10, the visual appearance of pork color (L*, a*, b*), sensory traits (color, marbling, firmness), pH value, WHC, and drip loss remained more or less similar in all groups.

Synbiotic-glyconutrient efficacy on the sensory traits and texture profile analysis (TPA) on finishing pig meat is shown in Table 4. The pork belly meat of the SB-GLN-treated group showed a tendency to increase (*p* < 0.1) tenderness and linearly increase (*p* < 0.05) in juiciness compared to the CON group. Moreover, the loin meat of the SB-GLN-treated group showed a linear increase (*p* < 0.05) in flavor. However, there were no differences observed on the texture and preference in pork belly and loin meat. The TPA parameter of hardness 1, cohesiveness, and gumminess and hardness 2, chewiness, and stringiness were higher (*p* < 0.05) in the SB-GLN group belly and loin meat, respectively. However, there were no differences observed on adhesiveness, fracture, and springiness Index.

Table 5, Table 6 and Table 7 illustrate the supplemental effect of synbiotic-glyconutrient on the fatty acid profile in pork belly fat, belly-lean meat, and loin-lean meat, respectively. The values of FA like butyric acid (C4:0), caproic acid (C6:0), undecylic acid (C11:0), tridecylic acid (C13:0), myristic acid (C14:0), pentadecyclic acid (C15:1), palmitic acid (C16:0, C16:1), margaric acid (C17:0, C17:1), stearic acid (C18:1,t; C18:1,c), eicosapentaenoic acid (C20:3n3; C20:4n6; C20:5n3), and lignoceric acid (C24:0) were significantly higher in the SB-GLN-treated group pork belly fat compared to CON (Table 5). Moreover, pigs that received SB-GLN exhibited a higher crude fat and lauric acid, myristic acid, pentacyclic acid, palmitic acid, margaric acid, Octadecanoic acid, Oleic acid, linoleic acid (C18:2n6c, LA), and Eicosapentaenoic acid (C20:5n3, EPA) profile in belly-lean meat (Table 6). Also, the FA profile of the SB-GLN-treated group loin-lean meat showed increased lauric acid (C12:0), myristic acid, palmitic acid, margaric acid, stearic acid, Oleic acid, linoleic acid, alpha-linoleic acid, and Eicosapentaenoic acid (Table 7). Furthermore, fatty acid content in pork belly fat, belly lean meat, and loin-lean meat of the SB-GLN-treated group showed increased docosahexaenoic acid (C22:6n3, DHA), nervonic acid (C24:1n9), omega 3, Σ saturated FA, Σ un-SFA, Σ mono-USFA, Σ poly-USFA, MUFA/SFA, and PUFA/SFA, and reduced omega 6, ω-6: ω-3 compared to CON (Table 5, Table 6 and Table 7).

## 4. Discussion

Genetic factors and rearing conditions such as nutrition are the key factors to determine the efficiency of pig farming and the production of high-quality pork [30]. In this regard, feed additives play a vital role, as they could improve the health of animals by utilizing nutrients and shaping their gut microbiome. The present study demonstrates that the inclusion of SB-GLN to finishing pigs’ diet linearly improved the body weight and increased the daily feed intake and daily weight gain. In line with this study, Chlebicz-Wójcik et al. [31] reported that symbiotic additives contribute to better weight gain in weaning-finishing pigs. On the other hand, Munezero et al. [32] stated that the inclusion of 0.5% SB-GLN to the finishing pig diet revealed better growth performance. Awad et al. [17] reported the dietary inclusion of synbiotics at the concentration of 1 g/kg showed improved body weight gain (BWG) and feed conversion ratio (FCR) in broilers. We presume that the reason for improved growth performance is mainly due to the effects of synbiotics, which helps the pigs to maintain beneficial microbial communities in their gut and improve feed digestion by altering bacterial metabolism. However, the previous literature reported the controversial impact of synbiotics on growth performance in monogastric animals. For example, Liong et al. [33] reported that the inclusion of a synbiotics (*Lactobacillus acidophilus* ATCC 4962, mannitol, fructooligosaccharide, and inulin) supplement to a high-fat or low-fat diet reveals no significant effects on the growth performance of growing pigs. Similarly, Erdogan et al. [34] reported that broilers fed a diet supplemented with 1 g/kg synbiotics had no effect on BWG and FCR. On the other hand, Hassanpour et al. [13] stated that broilers that received a 1 and 2 g/kg synbiotic additive showed improved daily weight gain without affecting feed efficacy. Furthermore, Cheng et al. [35] reported that the inclusion of a 1 g/kg symbiotic supplement (prebiotics-yeast cell wall and xylooligosaccharide; probiotics-*Clostridium butyricum, Bacillus licheniformis*, and *Bacillus subtilis*) to late-finishing pig’s diet had no improvement on their growth performance. These inconsistent findings could be partly explained by strain and number of microorganisms in the synbiotic components, survivability of the live organisms in the feed, prebiotic ingredients, inclusion level, or due to the dietary nutrient levels.

The quality of meat is usually determined by appearance and wholesomeness, and consequently influences the meat purchasing decision by consumers [36]. In the present study, pigs fed a diet supplemented with SB-GLN showed no changes in meat color in either SB-GLN or CON samples, which is in accordance with the results of Cheng et al. [35], who found similar effects in pigs fed a probiotic diet. The pH value of the meat is an important index to reflect the muscle contraction and glycolysis rate of pigs. Such pH values of the SB-GLN group meat ranged from 5.63 to 5.68% without significant differences observed between the CON groups. The current finding correlates with the results of Pieszka et al. [37]. The WHC of the meat is defined by the ability of fresh meat to retain its water during drip and cooking loss [38]. Additionally, drip loss and cooking loss greatly influence the juiciness of meat [39]. In the present study, the SB-GLN meat sample exhibited lower cooking loss than the CON group, which correlates with Liu et al. [40], who observed the reduced cooking loss of pork with dietary probiotics (yeasts, lactic acid-producing bacteria, and Bacillus subtilis). Similarly, Rybarczyk et al. [41] reported that the addition of the EM^®^ Bokashi probiotic to the pig’s diet resulted in a higher drip loss. The observed inconsistence in this result is probably due to the lesser physical resistance of immobilization of the water fraction which transverses the meat structural matrix and overcomes the fiber orientation to show variation in the drip loss [42].

Sensory evaluation is the result of scoring performed by trained panelists in order to provide the particular information on the acceptability or preference for one kind of meat [43,44]. Such sensory impression plays an important role in predicting the quality and purchasing decisions made by the consumers [45]. In 2004, Hansen et al. [46] reported that the sensory quality of pork was influenced by the manipulation of feed ingredients, including fructooligosaccharides, but in this study, finishing pigs that received a synbiotic glyconutrient supplement revealed a higher sensory profile of tenderness and juiciness (pork belly meat) and strong flavor (loin meat). On the other hand, Grela et al. [47] found less hardness of meat by dietary pre (inulin)-and probiotics (*Lactococcus lactis*, *Carnobacterium divergens*, *Lactobacillus casei Lactobacillus plantarum* and *Sacharomyces cerevisiae*). The reason for the increased tenderness and juiciness of belly and loin meat are probably due to the presence of fat which enhances WHC by lubricating the muscle fibers during cooking and increasing the tenderness of meat to reveal the sensation of juiciness [48]. Furthermore, the high flavor scores of the loin meat are likely due to rich flavor substances via lipid oxidative degradation. Previously, de Huidobro et al. [27] proved TPA as a viable method for the evaluation of the texture of various food items, with one advantage to assess multiple variables such as hardness, cohesiveness, springiness, and chewiness at one time. Such TPA parameters of hardness 1, cohesiveness, and gumminess were higher (*p* < 0.05) in the SB-GLN group belly, and hardness 2, chewiness, and springiness were higher (*p* < 0.05) in the loin meat. The higher values of hardness, chewiness, and gumminess of pork meat are likely due to the presence of myofibril proteins in meat, which cause a tougher network formation internally by enhancing the resistance to compression [49].

Pork lean meat is rich in PUFA due to the constant proportion of cell membrane phospholipids [50]. Previously, Chang et al. [51] reported that the pork meat of the probiotic-supplemented group (*Lactobacillus plantarum*) showed higher PUFA contents, with significantly higher levels of linolenic and linoleic acid, which corresponds with current findings. Additionally, Narayan et al. [52] reported that long chain *n*-3 FA possess health-promoting properties including EPA (C20:5*n*-3) and DHA (C22:6*n*-3), which were significantly higher in the SB-GLN-treated group meat compared to CON. Moreover, palmitic acid (C16:0) and stearic acid (C18:0) are considered to be predominant SFA in commercial pork meat [53]; such an FA profile was linearly increased in the SB-GLN-treated group pork belly fat, belly lean, and loin-lean meat. In 2012, Ross et al. [54] reported that meat samples of pigs that were supplemented with probiotics showed higher PUFA contents with significantly higher (*p* < 0.05) concentrations of linoleic acid (C18:2) and significantly lower monounsaturated FA; this partially agrees with the current findings. According to Pork Composition and Quality Assessment Procedures [55], the concentration of the linoleic acid and linolenic acid ratio should be around 5:1 to promote health and to reduce the risk of cardiovascular disease. Although the meat of the SB-GLN group showed higher linolenic and linoleic acid compared to the CON group, the *n*-3 and *n*-6 in the present study were also higher than recommended levels, indicating that this pork meat is well suited for human consumption.

## 5. Conclusions

Our study demonstrates that the inclusion of SB-GLN improved the growth performance of finishing pigs. Additionally, it enhanced the texture profile, sensory characteristics, and fatty acid profile of pork belly fat, belly lean, and lean loin meat. Based on this finding, it can be concluded that adding 0.5% of SB-GLN to the finishing pig diet would be more beneficial for the enhancement of the animal’s growth performance together with the increased nutritional quality of pork meat for human consumption.

## Figures and Tables

**Figure 1 foods-13-00105-f001:**
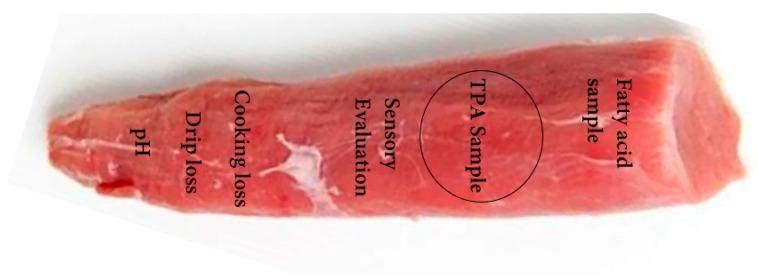
Schematic representation of the muscle sample for meat quality analysis.

**Figure 2 foods-13-00105-f002:**
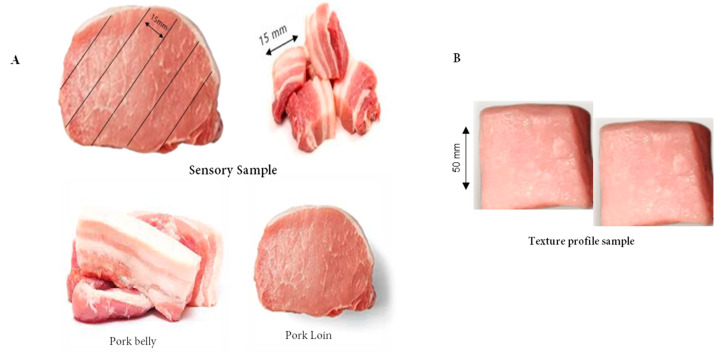
Schematic representation of the sample preparation for sensory evaluation (SE) and texture profile analysis. The SE and TPA samples for each measurement were cut parallel to the longitudinal orientation. Cubes were 15 mm and were measured perpendicular to the muscle fiber orientation according Ba et al. [28].

**Table 1 foods-13-00105-t001:** Basal diet composition (as-fed basis) of finishing pigs according to NRC regulation.

	Phase 1 (Week 0–5)	Phase 2 (Week 6–10)
**Raw Material**	**%**	**%**
Corn	63.7	68.9
Soybean meal	19.8	11.9
Rapeseed meal	3.00	4.00
Dried distillers’ grains soluble	5.00	7.00
Tallow	3.45	3.11
Molasses	2.00	2.00
Limestone	1.24	1.27
Mono-di-calcium phosphate	0.53	0.37
Salt	0.30	0.30
DL-Methonine	0.04	-
L-Lysine H_2_SO_4_	0.41	0.45
L-Threonine	0.06	0.07
L-Tryptophan (10%)	0.17	0.33
Vit/Min premix ^1^	0.20	0.20
Phytase	0.05	0.05
Carbohydrase	0.05	0.05
Total	100	100
**Calculated values**		
Moisture	12.9	13.0
Crude protein	16.7	14.4
Ether extract;	5.71	5.64
Fiber	2.95	2.89
Ash	5.07	4.72
Non-starch polysaccharides	120	116
Neutral detergent fiber	10.2	10.8
Acid-detergent fiber	2.98	3.09
Calcium	0.69	0.66
Phosphorus	0.42	0.38
Sodium	0.15	0.16
Chlorine	0.28	0.28
Potassium	0.83	0.71
Lysine	1.02	0.86
Methionine	0.32	0.26
Threonine	0.67	0.59
Tryptophan	0.19	0.18
Methonine + cystine	0.62	0.53

^1^ Provided per kg diet: Fe, 100 mg as ferrous sulfate; Cu, 17 mg as copper sulfate; Mn, 17 mg as manganese oxide; Zn, 100 mg as zinc oxide; I, 0.5 mg as potassium iodide; and Se, 0.3 mg as sodium selenite. Provided per kilograms of diet: vitamin A, 10,800 IU; vitamin D3, 4000 IU; vitamin E, 40 IU; vitamin K3, 4 mg; vitamin B1, 6 mg; vitamin B2, 12 mg; vitamin B6, 6 mg; vitamin B12, 0.05 mg; biotin, 0.2 mg; folic acid, 2 mg; niacin, 50 mg; D-calcium pantothenate, 25 mg.

**Table 2 foods-13-00105-t002:** Synbiotic-glyconutrient efficacy on finishing pig’s growth performance.

Traits	CON ^1^	TRT 1	TRT 2	SEM ^2^	*p* Value	
					Linear	Quadratic
**Body weight, kg**						
Initial	54.88	54.88	54.89	0.01	0.506	0.937
Week 5	81.14 ^b^	82.21 ^ab^	82.83 ^a^	0.47	0.045	0.705
Week 10	112.61 ^b^	115.99 ^ab^	117.02 ^a^	1.08	0.031	0.435
**Week 5**						
Average daily Gain, g	750 ^b^	781 ^ab^	798 ^a^	13.37	0.045	0.703
Average daily intake, g	2105 ^b^	2161 ^ab^	2199 ^a^	28.42	0.056	0.799
Gain to feed ratio	2.806	2.768	2.756	0.02	0.102	0.616
**Week 10**						
Average daily Gain, g	903 ^b^	965 ^ab^	977 ^a^	17.82	0.026	0.293
Average daily intake, g	2838	2940	2943	36.15	0.086	0.306
Gain to feed ratio	3.145	3.048	3.014	0.04	0.081	0.567
**Overall**						
Average daily Gain, g	827 ^b^	873 ^ab^	888 ^a^	15.35	0.031	0.432
Average daily intake, g	2471 ^b^	2551 ^ab^	2571 ^a^	28.94	0.051	0.438
Gain to feed ratio	2.991	2.923	2.898	0.03	0.066	0.532

^1^ Abbreviation: CON—Corn-soybean meal basal diet; TRT 1—CON+ 0.25% Synbiotic-glyconutrient (SB-GLN); and TRT 2—CON +0.5% SB-GLN. SEM ^2^-standard error of means. ^a, b^ means in the same row with different superscripts indicates significant (*p* value < 0.05).

**Table 3 foods-13-00105-t003:** Synbiotic-glyconutrient efficacy on finishing pig’s meat quality.

Traits	CON ^1^	TRT 1	TRT 2	SEM ^2^	*p* Value	
					Linear	Quadratic
**Week 10**						
Sensory Evaluation						
Color	3.13	3.31	3.28	0.13	0.426	0.510
Firmness	3.31	3.34	3.22	0.11	0.568	0.586
Marbling	3.22	3.06	3.31	0.10	0.549	0.158
Meat Color						
Lightness (L*)	51.97	51.95	51.82	0.39	0.791	0.906
Yellowness (a*)	14.53	14.66	14.73	0.10	0.192	0.762
Redness (b*)	5.89	6.05	5.90	0.13	0.979	0.361
LMA, mm^2^	7412.21 ^b^	7522.98 ^ab^	7557.57 ^a^	44.8	0.062	0.513
pH	5.56	5.63	5.68	0.06	0.206	0.854
WHC%	42.44	47.60	48.62	2.49	0.129	0.522
Drip loss, %						
d1	7.98	7.93	7.90	0.45	0.904	0.989
d3	13.28	13.08	12.99	0.29	0.504	0.882
d5	19.69	19.35	19.16	0.50	0.487	0.898
d7	24.83	24.19	23.68	0.42	0.101	0.906
Cooking loss, %	33.18 ^a^	31.98 ^ab^	30.96 ^b^	0.63	0.049	0.913

^1^ Abbreviation: CON—Corn-soybean meal basal diet; TRT 1—CON+ 0.25% Synbiotic-glyconutrient (SB-GLN); and TRT 2—CON + 0.5% SB-GLN. LMA—longissimus Muscle Area. WHC-Water Holding Capacity. SEM ^2^-standard error of means ^a, b^ means in the same row with different superscripts indicates significant (*p* value < 0.05).

**Table 4 foods-13-00105-t004:** Synbiotic-glyconutrient efficacy on sensory traits and texture profile on finishing pig meat.

Traits	CON ^1^	TRT 1	TRT 2	SEM ^2^	*p* Value	
Week 10					Linear	Quadratic
**Sensory traits**						
**Pork belly**						
Tenderness	3.19 ^ab^	3.81 ^b^	3.94 ^a^	0.26	0.068	0.460
Flavor	3.25	3.63	4.00	0.31	0.114	0.241
Juicy	3.38 ^b^	3.63 ^ab^	3.81 ^a^	0.29	0.053	0.497
Texture	3.13	3.50	3.63	0.28	0.238	0.727
Preference	3.25	3.56	3.75	0.32	0.284	0.874
**Loin**						
Tenderness	2.38	3.00	3.19	0.33	0.111	0.605
Flavor	2.63 ^b^	3.63 ^ab^	3.69 ^a^	0.29	0.027	0.213
Juicy	2.75	3.06	3.06	0.34	0.532	0.716
Texture	3.00	3.19	3.31	0.32	0.509	0.938
Preference	2.81	3.13	3.19	0.29	0.378	0.732
**Texture profile**						
**Pork belly**						
Hardness 1, N	136.67 ^b^	193.26 ^a^	201.98 ^a^	14.35	0.018	0.222
Hardness 2, N	116.75	123.40	163.19	21.58	0.178	0.554
Cohesiveness	0.59 ^b^	0.70 ^a^	0.73 ^a^	0.23	0.005	0.219
Adhesiveness, mm	2.51	2.67	2.96	0.28	0.306	0.849
Gumminess, N	80.09 ^b^	89.78 ^ab^	101.44 ^a^	7.12	0.078	0.914
Fracture, N	62.54	64.34	81.80	10.37	0.237	0.560
Stringiness, mm	6.29	10.38	11.27	2.12	0.149	0.561
Chewiness, N	50.46	62.43	83.30	18.34	0.252	0.844
Springiness Index, mm	0.83	0.87	0.90	0.06	0.424	0.865
**Loin**						
Hardness 1, N	125.75	168.64	170.19	26.77	0.285	0.552
Hardness 2, N	89.07 ^b^	103.68 ^ab^	140.75 ^a^	13.91	0.039	0.534
Cohesiveness	0.37	0.48	0.54	0.07	0.168	0.799
Adhesiveness, mm	1.08	1.87	1.95	0.38	0.157	0.484
Gumminess, N	62.52	75.56	83.28	12.54	0.286	0.868
Fracture, N	125.66	150.42	150.67	13.32	0.232	0.481
Stringiness, mm	6.81 ^b^	9.31 ^a^	9.93 ^a^	0.65	0.014	0.284
Chewiness, N	38.70 ^b^	67.17 ^ab^	79.63 ^a^	12.17	0.055	0.611
Springiness Index, mm	0.71	0.79	0.80	0.04	0.178	0.471

^1^ Abbreviation: CON—Corn-soybean meal basal diet; TRT 1—CON+ 0.25% Synbiotic-glyconutrient (SB-GLN); and TRT 2—CON + 0.5% SB-GLN. SEM^2^-standard error of means. ^a, b^ means in the same row with different superscripts indicates significant (*p* value < 0.05).

**Table 5 foods-13-00105-t005:** Synbiotic-glyconutrient efficacy on fatty acid profile in pork belly fat.

Fatty Acids	CON ^1^	TRT 1	TRT 2	SEM ^2^	*p* Value	
					Linear	Quadratic
C4:0	0.00	0.00	0.00			
C6:0	0.06 ^b^	0.11 ^a^	0.12 ^a^	0.016	0.043	0.266
C8:0	0.00 ^b^	0.02 ^ab^	0.03 ^a^	0.009	0.077	0.844
C10:0	0.01	0.03	0.04	0.02	0.202	0.855
C11:0	0.00	0.00	0.00	-		
C12:0	0.07 ^b^	0.13 ^ab^	0.16 ^a^	0.022	0.031	0.611
C13:0	0.00	0.00	0.00			
C14:0	0.99 ^b^	1.47 ^ab^	1.60 a	0.17	0.052	0.445
C14:1	0.08 ^b^	0.23 ^a^	0.21 ^a^	0.0455	0.081	0.178
C15:0	0.06 ^b^	0.22 ^ab^	0.20 ^a^	0.048	0.086	0.150
C15:1	0.00	0.01	0.02	0.01	0.110	0.882
C16:0	28.47 ^a^	16.72 ^ab^	12.97 ^a^	1.29	0.0001	0.045
C16:1	2.49 ^b^	2.75 ^ab^	2.80 ^a^	0.09	0.067	0.448
C17:0	0.38 ^b^	0.50 ^ab^	0.55 ^a^	0.05	0.086	0.665
C17:1	0.20	0.28	0.34	0.05	0.094	0.896
C18:0	8.53 ^b^	10.51 ^ab^	12.21 ^a^	1.23	0.079	0.926
C18:1,t	3.54	3.55	3.64	0.33	0.833	0.926
C18:1,c	42.19 ^b^	45.63 ^ab^	46.65 ^a^	0.624	0.002	0.163
C18:2n6t	0.00 ^b^	0.02 ^ab^	0.04 ^a^	0.009	0.029	0.753
C18:2n6c, LA	10.14	13.12	13.68	1.45	0.136	0.522
C18:3n6	0.00	0.03	0.02	0.01	0.153	0.207
C18:3n3, ALA	0.40 ^b^	0.52 ^ab^	0.56 ^a^	0.052	0.073	0.603
C20:0	1.33	1.96	1.65	0.33	0.514	0.286
C20:1	0.01	0.00	0.01	0.01	0.745	0.218
C20:2	0.31	0.29	0.27	0.07	0.717	0.958
C20:3n6	0.04	0.05	0.06	0.01	0.236	0.884
C21:0	0.09	0.06	0.06	0.03	0.630	0.822
C20:3n3	0.00	0.00	0.02	0.01	0.266	0.506
C20:4n6	0.03	0.03	0.03	0.01	1.000	0.718
C20:5n3, EPA	0.06 ^b^	0.13 ^a^	0.15 ^a^	0.011	0.001	0.131
C22:0	0.12 ^b^	0.37 ^a^	0.38 ^a^	0.038	0.003	0.044
C22:1n9	0.00 ^b^	0.04 ^ab^	0.06 ^a^	0.008	0.166	0.665
C22:2	0.01	0.04	0.04	0.01	0.121	0.418
C23:0	0.03	0.04	0.05	0.01	0.166	0.665
C24:0	0.00	0.01	0.00			
C22:6n3, DHA	0.10 ^b^	0.42 ^a^	0.55 ^a^	0.061	0.002	0.256
C24:1n9	0.30 ^b^	0.75 ^ab^	0.84 ^a^	0.16	0.055	0.403
ω-3 fatty acid	0.56 ^b^	1.06 ^a^	1.26 ^a^	0.036	<.0001	0.013
ω-6 fatty acid	10.21	13.24	13.83	1.45	0.129	0.519
ω-6: ω-3	18.14	12.71	11.07	1.98	0.045	0.465
Σ Saturated FA	40.12	32.14	30.00	1.91	0.009	0.259
Σ USFA	59.88	67.86	70.00	1.91	0.009	0.259
Σ mono-USFA	48.80 ^b^	53.24 ^ab^	54.58 ^a^	0.69	0.001	0.120
Σ Poly-USFA	11.09 ^b^	14.63 ^ab^	15.42 ^a^	1.5	0.087	0.484
MUFA/SFA	1.24 ^b^	1.68 ^ab^	1.82 ^a^	0.09	0.004	0.256
PUFA/SFA	0.29	0.47	0.52	0.07	0.051	0.426
Total FA	100.00	100.00	100.00	-	-	-

^1^ Abbreviation: CON—Corn-soybean meal basal diet; TRT 1—CON+ 0.25% Synbiotic-glyconutrient (SB-GLN); and TRT 2—CON + 0.5% SB-GLN. SEM^2^-standard error of means. ^a, b^ means in the same row with different superscripts indicates significant (*p* value < 0.05). Fatty acids: C10:0 (Capric acid); C12:0 (Lauric acid); C14:0 (Myristic acid); C14:1 (Myristoleic acid); C15:0 (Pentadecylic acid); C16:0 (Palmitic acid); C16:1 *n*-9 (Cis-7 Hexadecenoic acid); C16:1 *n*-7 (Palmitoleic acid); C17:0 (Margaric acid); C17:1 (Heptadecenoic acid); C18:0 (Stearic acid); C18:1 isomer (Octadecenoic acid isomer); C18:1 *n*-9 (Oleic acid); C18:1 *cis*-11 (Vaccenic acid); C18:2 *n*-6 (Linoleic acid); C18:3 *n*-6 (γ-linolenic acid); C18:3 *n*-3 (α-linolenic acid); C20:0 (Arachidic acid); C20:1 (Gadoleic acid); C20:2 *n*-6 (Eicosadienoic acid); C20:3 *n*-6 (Dihomo-γ-linolenic acid); C20:4 *n*-6 (Arachidonic acid); C20:3 *n*-3 (Eicosatrienoic acid); C22:2 *n*-6 (Docosadienoic acid); C20:5 *n*-3 (Eicosapentaenoic acid); C24:0 (Lignoceric acid); C22:5 *n*-3 (Docosapentaenoic acid); C22:6 *n*-3 (Docosahexaenoic acid).

**Table 6 foods-13-00105-t006:** Synbiotic-glyconutrient efficacy on fatty acid profile in pork belly-lean meat.

Fatty Acids	CON ^1^	TRT 1	TRT 2	SEM ^2^	*p* Value	
					Linear	Quadratic
Crude fat, %	42.22 ^b^	49.70 ^a^	52.64 ^a^	1.67	0.004	0.312
C4:0	0.00	0.00	0.00	-	-	-
C6:0	0.06 ^b^	0.15 ^a^	0.15 ^a^	0.015	0.005	0.083
C8:0	0.04	0.05	0.04	0.01	0.776	0.625
C10:0	0.04	0.05	0.05	0.01	0.490	0.686
C11:0	0.00	0.00	0.00	-	-	-
C12:0	0.07 ^b^	0.10 ^a^	0.11 ^a^	0.006	0.002	0.152
C13:0	0.00	0.02	0.02	0.01	0.188	0.544
C14:0	1.28 ^b^	1.63 ^a^	1.67 ^a^	0.08	0.017	0.166
C14:1	0.06 ^b^	0.20 ^a^	0.16 ^a^	0.01	0.003	0.003
C15:0	0.13 ^b^	0.34 ^ab^	0.40 ^a^	0.07	0.054	0.454
C15:1	0.00 ^b^	0.03 ^a^	0.04 ^a^	0.008	0.028	0.506
C16:0	25.53 ^a^	19.68 ^ab^	16.58 ^b^	0.71	0.0001	0.168
C16:1	2.67	2.49	2.68	0.13	0.947	0.293
C17:0	0.35	0.55	0.60	0.08	0.091	0.515
C17:1	0.17 ^b^	0.29 ^ab^	0.33 ^a^	0.04	0.049	0.498
C18:0	10.22 ^b^	11.75 ^ab^	12.39 ^a^	0.3	0.002	0.273
C18:1,t	3.47	3.41	3.57	0.12	0.605	0.487
C18:1,c	43.53 ^b^	44.64 ^ab^	46.16 ^a^	0.45	0.006	0.727
C18:2n6t	0.01	0.02	0.03	0.01	0.307	0.837
C18:2n6c, LA	8.69 ^b^	10.12 ^a^	10.38 ^a^	0.37	0.019	0.258
C18:3n6	0.03	0.04	0.04	0.02	0.583	0.749
C18:3n3, ALA	0.50	0.59	0.61	0.04	0.142	0.489
C20:0	2.13	2.18	2.13	0.13	1.000	0.782
C20:1	0.01	0.00	0.00	0.01	0.266	0.506
C20:2	0.28	0.24	0.24	0.01	0.137	0.314
C20:3n6	0.05	0.05	0.05	0.01	0.323	0.557
C21:0	0.17	0.12	0.15	0.03	0.626	0.318
C20:3n3	0.00	0.00	0.00	-	-	-
C20:4n6	0.04	0.03	0.04	0.01	0.652	0.219
C20:5n3, EPA	0.00 ^b^	0.12 ^a^	0.14 ^a^	0.01	0.0001	0.009
C22:0	0.03 ^b^	0.22 ^ab^	0.29 ^a^	0.055	0.016	0.395
C22:1n9	0.00	0.06	0.05	0.01	0.151	0.762
C22:2	0.03	0.05	0.03	0.01	1.000	0.135
C23:0	0.07	0.06	0.05	0.01	0.151	0.762
C24:0	0.02	0.01	0.01	0.01	0.567	0.582
C22:6n3, DHA	0.10 ^b^	0.39 ^a^	0.49 ^a^	0.058	0.003	0.224
C24:1n9	0.25 ^b^	0.37 ^a^	0.38 ^a^	0.02	0.004	0.045
ω-3 fatty acid	0.60 ^b^	1.10 ^a^	1.23 ^a^	0.029	<.0001	0.002
ω-6 fatty acid	8.82 ^b^	10.25 ^ab^	10.53 ^a^	0.39	0.021	0.277
ω-6: ω-3	14.67	9.40	8.61	0.70	0.001	0.041
Σ Saturated FA	40.12 ^a^	36.89 ^ab^	34.63 ^b^	0.66	0.001	0.573
Σ Un-SFA	59.89	63.11	65.37	0.66	0.001	0.575
Σ mono-USFA	50.16 ^b^	51.48 ^ab^	53.35 ^a^	0.51	0.004	0.683
Σ Poly-USFA	9.73	11.63	12.02	0.39	0.006	0.170
MUFA/SFA	1.25 ^b^	1.40 ^ab^	1.54 ^a^	0.03	0.002	0.960
PUFA/SFA	0.24 ^b^	0.32 ^a^	0.35 ^a^	0.02	0.003	0.348
Total FA	100.00	100.00	100.00	-	-	-

^1^ Abbreviation: CON—Corn-soybean meal basal diet; TRT 1—CON+ 0.25% Synbiotic-glyconutrient (SB-GLN); and TRT 2—CON + 0.5% SB-GLN. SEM^2^-standard error of means. ^a, b^ means in the same row with different superscripts indicates significant (*p* value < 0.05). Fatty acids: C10:0 (Capric acid); C12:0 (Lauric acid); C14:0 (Myristic acid); C14:1 (Myristoleic acid); C15:0 (Pentadecylic acid); C16:0 (Palmitic acid); C16:1 *n*-9 (Cis-7 Hexadecenoic acid); C16:1 *n*-7 (Palmitoleic acid); C17:0 (Margaric acid); C17:1 (Heptadecenoic acid); C18:0 (Stearic acid); C18:1 isomer (Octadecenoic acid isomer); C18:1 *n*-9 (Oleic acid); C18:1 *cis*-11 (Vaccenic acid); C18:2 *n*-6 (Linoleic acid); C18:3 *n*-6 (γ-linolenic acid); C18:3 *n*-3 (α-linolenic acid); C20:0 (Arachidic acid); C20:1 (Gadoleic acid); C20:2 *n*-6 (Eicosadienoic acid); C20:3 *n*-6 (Dihomo-γ-linolenic acid); C20:4 *n*-6 (Arachidonic acid); C20:3 *n*-3 (Eicosatrienoic acid); C22:2 *n*-6 (Docosadienoic acid); C20:5 *n*-3 (Eicosapentaenoic acid); C24:0 (Lignoceric acid); C22:5 *n*-3 (Docosapentaenoic acid); C22:6 *n*-3 (Docosahexaenoic acid).

**Table 7 foods-13-00105-t007:** Synbiotic-glyconutrient efficacy on fatty acid profile in pork loin-lean meat.

Fatty Acids	CON ^1^	TRT 1	TRT 2	SEM	*p* Value	
					Linear	Quadratic
Crude fat, %	5.73 ^b^	6.24 ^a^	6.66 ^a^	0.18	0.010	0.867
C4:0	0.00	0.00	0.00	-	-	-
C6:0	0.00	0.00	0.00	-	-	-
C8:0	0.00	0.00	0.00	-	-	-
C10:0	0.0	0.00	0.02	0.011	0.266	0.506
C11:0	0.00	0.00	0.00			
C12:0	0.13 ^b^	0.17 ^a^	0.19 ^a^	0.011	0.008	0.360
C13:0	0.00	0.00	0.00			
C14:0	1.09 ^b^	1.14 ^ab^	1.17 ^a^	0.013	0.009	0.523
C14:1	0.00	0.00	0.00	-	-	-
C15:0	0.00	0.00	0.00	-	-	-
C15:1	0.00	0.00	0.00	-	-	-
C16:0	30.16 ^a^	20.51 ^a^	13.77 ^b^	2.1	0.002	0.593
C16:1	2.90 ^b^	3.25 ^a^	3.39 ^a^	0.06	0.002	0.204
C17:0	0.29 ^b^	0.48 ^a^	0.50 ^a^	0.03	0.003	0.072
C17:1	0.10	0.24	0.27	0.058	0.090	0.524
C18:0	10.85 ^a^	14.46 ^a^	14.72 ^a^	1	0.034	0.221
C18:1, t	0.00	0.00	0.01	0.002	0.266	0.506
C18:1, c	44.35 ^b^	48.35 ^a^	54.22 ^a^	2.19	0.018	0.740
C18:2 n6	0.06 ^b^	0.08 ^ab^	0.09 ^a^	0.005	0.011	0.343
C18:2 n6c, LA	7.37	7.90	8.07	0.65	0.479	0.825
C18:3 n6	0.00	0.00	0.00	-	-	-
C18:3 n3, ALA	0.53 ^b^	0.91 ^ab^	0.97 ^a^	0.12	0.042	0.314
C20:0	0.24	0.28	0.29	0.048	0.561	0.826
C20:1	1.06	1.20	1.27	0.08	0.115	0.718
C20:2	0.30	0.31	0.32	0.054	0.804	0.914
C20:3n6	0.06	0.09	0.08	0.012	0.180	0.277
C21:0	0.00	0.00	0.00	-	-	-
C20:3n3	0.48	0.50	0.52	0.014	0.099	0.892
C20:4n6	0.00	0.00	0.00	-	-	-
C20:5n3, EPA	0.00 ^b^	0.04 ^ab^	0.06 ^a^	0.012	0.012	0.813
C22:0	0.00	0.00	0.00	-	-	-
C22:1n9	0.01	0.03	0.04	0.013	0.210	0.312
C22:2	0.00	0.00	0.00	-	-	-
C23:0	0.05	0.08	0.08	0.013	0.210	0.312
C24:0	0.00	0.00	0.00	-	-	-
C22:6n3, DHA	0.00	0.00	0.00	-	-	-
C24:1n9	0.00	0.00	0.00	-	-	-
ω-3 fatty acid	0.53 ^b^	0.95 ^ab^	1.03 ^a^	0.12	0.025	0.303
ω-6 fatty acid	7.48	8.07	8.24	0.66	0.448	0.803
ω-6: ω-3	15.94 ^a^	8.64 ^ab^	8.53 ^b^	2.35	0.067	0.258
Σ Saturated FA	42.81	37.11	30.71	2.16	0.007	0.900
Σ Un-SFA	57.19 ^b^	62.90 ^ab^	69.29 ^a^	2.16	0.006	0.900
Σ mono-USFA	48.41 ^b^	53.07 ^ab^	59.19 ^a^	2.25	0.014	0.800
Σ Poly-USFA	8.78	9.83	10.10	0.65	0.206	0.652
MUFA/SFA	1.14 ^b^	1.48 ^ab^	1.94 ^a^	0.17	0.016	0.823
PUFA/SFA	0.21 ^b^	0.27 ^ab^	0.33 ^a^	0.026	0.017	0.881
Total FA	100.00	100.00	100.00	-	-	-

^1^ Abbreviation: CON—Corn-soybean meal basal diet; TRT 1—CON+ 0.25% Synbiotic-glyconutrient (SB-GLN); and TRT 2—CON + 0.5% SB-GLN. SEM^2^-standard error of means. ^a, b^ means in the same row with different superscripts indicates significant (*p* value < 0.05). Fatty acids: C10:0 (Capric acid); C12:0 (Lauric acid); C14:0 (Myristic acid); C14:1 (Myristoleic acid); C15:0 (Pentadecylic acid); C16:0 (Palmitic acid); C16:1 *n*-9 (Cis-7 Hexadecenoic acid); C16:1 *n*-7 (Palmitoleic acid); C17:0 (Margaric acid); C17:1 (Heptadecenoic acid); C18:0 (Stearic acid); C18:1 isomer (Octadecenoic acid isomer); C18:1 *n*-9 (Oleic acid); C18:1 *cis*-11 (Vaccenic acid); C18:2 *n*-6 (Linoleic acid); C18:3 *n*-6 (γ-linolenic acid); C18:3 *n*-3 (α-linolenic acid); C20:0 (Arachidic acid); C20:1 (Gadoleic acid); C20:2 *n*-6 (Eicosadienoic acid); C20:3 *n*-6 (Dihomo-γ-linolenic acid); C20:4 *n*-6 (Arachidonic acid); C20:3 *n*-3 (Eicosatrienoic acid); C22:2 *n*-6 (Docosadienoic acid); C20:5 *n*-3 (Eicosapentaenoic acid); C24:0 (Lignoceric acid); C22:5 *n*-3 (Docosapentaenoic acid); C22:6 *n*-3 (Docosahexaenoic acid).

## Data Availability

The data presented in this study are available on request from the corresponding author. The data are not publicly available due to ethical restriction.

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
