# Peer review of "Synbiotic-Glyconutrient Additive Reveals a Conducive Effect on Growth Performance, Fatty Acid Profile, Sensory Characteristics, and Texture Profile Analysis in Finishing Pig"

_foods, 2023, doi:10.3390/foods13010105_

Round 1
Reviewer 1 Report
Comments and Suggestions for Authors
Please see the attachment

Author Response
Response to reviewer 1
Author would like to thank the reviewer for spending his precious time to review our paper and making positive feed-back towards our manuscript. We have tried to modified the manuscript according to your comments using a red-color text.
Please rephrase this to something like: during lairage time animals were fasted with ad libitum access to water...
Response: Author would like to thank the reviewer for pin pointing this valid comment, which help us improve the quality of the manuscript. We have revised the sentence according to your comments.
what about the stunning method? Please describe in the text. Also, what is exsanguination and evisceration method?
Response: Sorry for the typographical error. As per Korean government rules we killed the pigs using stunning method. We have revised it.
please replace by after slaughter/post mortem or similar
Response: Thank you so much we have revised the sentence as per your comments.
does this mean that you took only 4 LD muscles for colour, pH, WHC etc? Note that this is very low number for any kind of statistical analysis.
Response:
please put the name and country of producer
Response: As per reviewer suggestion we have include the name and country of producer for Konica Minolta CR-400 Chroma meter instrument.
what were the units that you use for describing WHC?
Response: The WHC of meat affects its characteristics. Besides it is a crucial criterion for assessing quality of meat. Psi (Pounds per square inch)- is the units used for describing WHC. In this study we used filter-paper press method. First, we placed the meat sample between the folded filter paper (centre). Then the filter papers and meat sample were kept under rigid flat surface of glass plate. Then we apply pressure (40 psi) on it for 5 minutes. Remove the weight, separate the meat flake from the filter papers. Areas of the compressed sample and the expressed humidity were defined and determined by using a digitalized area-line sensor (MT-10S, M.T. Precision Co. Ltd., Tokyo, Japan). The ratio of water: meat area was then calculated, giving a measure of WHC (a smaller ratio indicates increased WHC. Finally, WHC was calculated using the following formula= [(initial sample – compressed sample)/initial weight] ×100.
why did you perform analysis for 7 consecutive days? Please explain and reference.
Response: Drip loss, as an indicator of the meat WHC, is one of the important parameters for both the meat industry and the consumer to evaluate meat quality. For the meat industry, drip loss of meat is known to influence its technological quality (such as processing yield) and economic benefits. For the consumer, higher drip loss reduces the tenderness, juiciness and sensory quality of the meat, causing lower consumer acceptance. Actually, we follow Honikel et al. (1998) method to determine the drip loss.
Reference: Honikel, K.O. Reference methods for the assessment of physical characteristic of meat. Meat Sci. 1998, 49, 447–457.
why 15 mm? Please can you reference this?
Response: Apologize for typographical error. Actually, we use 15mm (1.5 cm ) sample for sensory evaluation (Choe et al. 2016), for TPA analysis we took 30 mm cube sample.
Ref: Choe JH, Choi MH, Rhee MS, Kim BC. Estimation of sensory pork loin tenderness using Warner-Bratzler shear force and texture profile analysis measurements. Asian-Australasian journal of animal sciences. 2016 Jul;29(7):1029.
if I see correctly that would be 8 slices of 15 mm thick chops per one animal? After this you stated that you used cubes. Now, I am little confused...why so many slices? Did you not cut one sample into smaller subsamples (cubes; what size?)
Response: Apologize for this writing mistake. Actually, we took one big sample/ trt and cut them into smaller subsamples (sensory samples- 1.5 cm; TPA cube samples- 3 cm).
Ref: Choe JH, Choi MH, Rhee MS, Kim BC. Estimation of sensory pork loin tenderness using Warner-Bratzler shear force and texture profile analysis measurements. Asian-Australasian journal of animal sciences. 2016 Jul;29(7):1029.
Here and also in the tables and results and discussion it would be beneficial to incorporate the measure of atherogenic potential of fatty acid (IA). For calculation see Chen, J.; Liu, H. Nutritional Indices for Assessing Fatty Acids: A Mini-Review. Int. J. Mol. Sci. 2020,
Response: Author would apologize for denying reviewer suggestion. Actually, our research team wants to current method. For sure we will implement your valid suggestion in the future work.
This whole section is very scarcly described. I would prefer if you put the table and after or just before the table to describe your results.
Response: We have revised this section as per your comments.
please put at the end of the table meaning of Phase 1 and phase 2. The tables should be self-explanatory
Response: We have included the information for phase 1 and phase 2, accordingly.
But this was not the case in this study. this part should also be incorporated into text
this sentence could be rephrased
Response: Thank you so much, we have revised the sentence as per your comments.
I would say that processors use the L,a,b to determine meat colour. Consumers just evaluate it by eye.
Response: We have revised the sentence accordingly.
which prebiotis and probiotics?
Response: Prebiotic (inulin) and probiotic (Lactococcus lactis, Carnobacterium divergens, Lactobacillus casei Lactobacillus plantarum and Sacharomyces cerevisiae) were used in Grela et al. study.
This can be a bit longer with discussion of implications
Response: we have revised the conclusion section as per your comments.
Reviewer 2 Report
Comments and Suggestions for Authors
The manuscript tried to investigate the effect of symbiotic and glyconutrients on meat quality. Generally, there are quite a lot of points that need to be addressed. Please insert the line number, otherwise, it is hard for reviewers to point out the problems.
Introduction: there are limited introductions on the effects of SB-GLN on other research works and the novelty of your research should be clarified. Furthermore, the authors mentioned that "So far numerous studies addressed the effect of symbiotic and glyconutrients (SB-GLN) in ruminants’ performance and meat quality", but there are neither references nor citations. There are several similar problems in other parts of the manuscript, please add the references accordingly.
Material and method: the section on "Meat Quality" is not clear. It should be clarified more clearly and restructured like pH, WHC, color (subsection), and the references the author referred to.
There are too many tables, please summarize them.
Table 4 is out of range, some parts can not be read.
For the TPA, it is unsure if the authors use cooked meat or raw meat. According to Figure 1, it seems that the authors carried out all the measurements via one thick chop. Usually, the sample should be obtained from the center of the meat by a hole puncher.
Tables 2-8: what happens to other values that do not have any markers?
It is better to combine the results and discussion so that the reader can easily follow and avoid the repetition
A conclusion is missing.
Comments on the Quality of English Language
It is better to paper proof by a native English speaker
Author Response
Response to reviewer 2
The manuscript tried to investigate the effect of symbiotic and glyconutrients on meat quality. Generally, there are quite a lot of points that need to be addressed. Please insert the line number, otherwise, it is hard for reviewers to point out the problems.
Response: Actually, we insert the line number while submission. But when we receive the revision file, the line no is missing. Apologize for this basic mistake. As per your comments we have include the line numbers. Thank you so much for giving us a chance to revise the manuscript.
Introduction: there are limited introductions on the effects of SB-GLN on other research works and the novelty of your research should be clarified. Furthermore, the authors mentioned that "So far numerous studies addressed the effect of symbiotic and glyconutrients (SB-GLN) in ruminants’ performance and meat quality", but there are neither references nor citations. There are several similar problems in other parts of the manuscript, please add the references accordingly.
Response: Thank you so much for pointing this. We have revised the introduction accordingly.
Material and method: the section on "Meat Quality" is not clear. It should be clarified more clearly and restructured like pH, WHC, color (subsection), and the references the author referred to.
Response: Author would like to thank the reviewer for pointing this valid comment. We have included the reference accordingly.
There are too many tables, please summarize them.
Response: Thank you so much. We have summarized all the tables accordingly.
For the TPA, it is unsure if the authors use cooked meat or raw meat. According to Figure 1, it seems that the authors carried out all the measurements via one thick chop. Usually, the sample should be obtained from the center of the meat by a hole puncher.
Response: We use raw meat for TPA analysis. Yes, the reviewer point of view is correct. Actually, we use took one big size sample and cut the center portion for (5 cm) TPA analysis. We have corrected the fig 1. Accordingly.
Tables 2-8: what happens to other values that do not have any markers? It is better to combine the results and discussion so that the reader can easily follow and avoid the repetition
Response: Author would like to thank the reviewer for pointing this valid comment. Apologize for denying this suggestion, actually our research team wish to keep the result and discussion section separate. We will follow this valid comment in future research work.
A conclusion is missing.
Response: Thank you so much for pointing this. We have included it accordingly.
Reviewer 3 Report
Comments and Suggestions for Authors
The article presents interesting data on growth performance, fatty acid profile and pork meat obtained with the addition of symbiotic glyconutrients.
Unfortunately, the manuscript is poorly formatted. The contents of the tables 4,5,6,7,8 are presented in such a way that it is not possible to fully understand what is presented in the first columns. Standard errors of average values are not presented, although the footnote says that these values are in the tables.
I believe that the manuscript should be corrected and submitted again.
- The authors should redo the tables 4,5,6,7,8. The first columns of these tables should be inserted into the text of the article in such a way that they are visible. The tables do not have columns showing standard errors, although the footnote says that these values are in the tables. The standard errors must be inserted.
- Yes.
- The topic is original.
- The article is not well written because it is impossible to get data from the tables 4,5,6,7,8.
- It is impossible to judge the conclusions because the data in the tables is not visible.
- It's hard to say until the data in the tables is fully visible.
Author Response
The article presents interesting data on growth performance, fatty acid profile and pork meat obtained with the addition of symbiotic glyconutrients.
Unfortunately, the manuscript is poorly formatted. The contents of the tables 4,5,6,7,8 are presented in such a way that it is not possible to fully understand what is presented in the first columns. Standard errors of average values are not presented, although the footnote says that these values are in the tables. I believe that the manuscript should be corrected and submitted again.
Response: Author would thank the reviewer for giving us a chance to do revision and re-submit the manuscript. As per your comments, we have align the tables and change the format.
Round 2
Reviewer 1 Report
Comments and Suggestions for Authors
Please see the attachment.

Author Response
Author would like to thank the reviewer for spending his precious time to review our paper and giving us a chance to revise. We have tried to modified the manuscript according to your comments using a red-color text.
- Line no: 153; This is extremely small amount of the sample. Are you sure that this is the correct number?
Response: Author would like to thank the reviewer for pin pointing this valid comment. Apologize for typographical error. It was 4.0 g (4 gram) and not 0.4 g. We have revised it accordingly.
- Line no: 163; Please add information on the size of cubes, and the number of subsamples per biological sample.
Response: Thank you so much we have mention it accordingly.
- Line no: 259; Is there tendency (trend) when comparing between groups? (please look at L253) If yes, please provide this information with subscript letters in the table. This should be also checked for other tables.
Response: Author would like to thank the reviewer for raising this valid comment. We have revised the tables accordingly.
- Table 3; Is there a difference (or tendency) between those figures? The absolute difference is quite big.
Response: Thank you so much for pin pointing this comment. Actually, there was no difference observed.
- WHC: Are you sure that there is no difference?
Response: Thank you so much for this comment. Yes, we crosscheck the data and there was no difference observed.
- Drip loss: Is there a difference? Or 9.35 is actually 19.35?
Response: Author would like to thank the reviewer for pin pointing this comment. Apologize for typographical error. It was 19.35 and not 9.35 g. We have revised it accordingly.
- Table 4, Please check again the table for differences or tendencies (p<0.1). There are quite big differences in absolute values, so I presume that there are some differences between them.
Response: As per reviewer comments we have crosscheck the data and there was no difference observed.
Reviewer 2 Report
Comments and Suggestions for Authors
The authors still do not address my comments. The next time, when you answer the reviewer, please mention with line number to show where the answer corresponds to the reviewer's comments. As I mentioned before
"Tables 2-8: what happens to other values that do not have any markers?", the authors did not answer.
The novelty of the study should be clarified.
There are also no improvements to the tables, please try to combine some tables to reduce the number of the tables.
Line 288: it should be "Tables 6,7, and 8 illustrate... "
I still insist the authors should combine the results and discussion for better understanding.
The conclusion is too general and should contain the main finding. In order to keep the manuscript to be objective. The subjective words like "we", "I" should be avoided and it is recommended to use the passive tense.
Comments on the Quality of English LanguageLine 288: it should be "Tables 6,7, and 8 illustrate...
English should be paper-proof by a native English speaker or an agency.
Author Response
The authors still do not address my comments. The next time, when you answer the reviewer, please mention with line number to show where the answer corresponds to the reviewer's comments. As I mentioned before
Response: Author would like to thank the reviewer for giving us a chance to reconsider our work. Authors really felt sorry for not mentioning the response with line numbers.
"Tables 2-8: what happens to other values that do not have any markers?", the authors did not answer.
Response: Apologize. Usually, we put a,b markers only for statistical significant values. We realize the importance of markers from your valid comment. We try to follow and use the letter indexes for statistically insignificant values too. We have revised the tables 2-8 accordingly.
The novelty of the study should be clarified.
Response: Probiotics, a “live microbial feed additive” are known to improve the performance of the host by improving their gut microbial balance. The combination of pre-and probiotic in the form of synergism is known as “synbiotics”, contains a substrate has been selectively utilized by co-administered microorganism. The primary reason for using synbiotic in food products, is that probiotics do not survive well in the digestive tract without prebiotic foods. Without the necessary food source for probiotic bacteria, their tolerance to temperature, oxygen and pH level may decreases. Glyconutrients (plant sugars) are rich in anti- inflammatory and antimicrobials properties. It can increase the energy efficiency and health of the host and promote their cellular integrity.
In 2020, De Vries et al. noted that inclusion of yeast cell wall ß-Glucans in has significantly increase the gut health of pigs. Similarly, Awad et al. (2009) reported dietary inclusion of synbiotic at the concentration of 1 g/kg had improved the body weight gain and feed efficacy in broilers. On the other hand, Lee et al. (2009) noted that dietary inclusion of synbiotic containing a probiotic originating from anaerobic microbiota (bacteria—109 CFU/ml, yeast—105 CFU/ml, molds—103 CFU/ml) and a prebiotic (MOS, sodium acetate, ammonia citrate) has improved digestion of nutrients in weaning pigs.
To date, lots of studies have been demonstrated the effect of prebiotic and probiotic (alone), and symbiotic (combination) efficacy in animals with better performance but study on the combination of symbiotic and glyconutrients in non-ruminants is not well elucidated.
In 2017, Valencia et al. [21] reported that combination of probiotic-glyconutrient has increased live weight gain and decreased the mortality rate and lower the non-esterified fatty acids in Holstein calves. Similarly, Castro-Perez et al. [22] noted that dietary SB-GLN improved the growth perfor-mance and carcass weight in lambs.
The above-mentioned literatures have provoked us to hypothesize and initiate this research to know whether the inclusion of SB-GLN combination could enhance the growth performance, fatty acid profile, and meat quality of pigs or not.
Thus, we conduct this novel research on finishing pig to examine the growth performance, fatty acid profile, and the quality of pork meat by adding SB-GLN additive to their diet.
There are also no improvements to the tables, please try to combine some tables to reduce the number of the tables.
Response: Thank you so much for your valid comment. We have combined table 4 and 5 as per your comments.
Line 288: it should be "Tables 6,7, and 8 illustrate... "
Response: Author would like to thank the reviewer for pointing error which helped us to improve the quality of the manuscript. We have revised the Line no 278, accordingly.
I still insist the authors should combine the results and discussion for better understanding.
Response: Apologize. As a first author I agree to combine the result and discussion. But correspondence and co-author still wish to keep it separate. Also, other two reviewer asked us to keep it separate section. I beg you for the understanding. I assure you that I will follow this valid comment in the upcoming article.
The conclusion is too general and should contain the main finding. In order to keep the manuscript to be objective. The subjective words like "we", "I" should be avoided and it is recommended to use the passive tense.
Response: Thank you so much for pinpointing this valid comment. We have revised the conclusion
Reviewer 3 Report
Comments and Suggestions for Authors
Basically, the authors took into account the comments of the reviewers.
However, the authors should correct the tables.
Table 1. Authors need to keep not more than 3 digits. This is the maximum accuracy of this kind of determinations. For example, the first row should be:
|
Corn |
63.7 |
68.9 |
Tables 2-8. Based on the standard errors provided, all values not marked with letter indexes are statistically insignificant. They must be marked with the same letter index. For example (Table 2, first row):
|
Initial |
54.88a |
54.88a |
54.89a |
Author Response
Response to reviewer 3
Basically, the authors took into account the comments of the reviewers. However, the authors should correct the tables.
Response: Author would like to thank the reviewer for giving us a chance to reconsider our work.
Table 1. Authors need to keep not more than 3 digits. This is the maximum accuracy of this kind of determinations. For example, the first row should be:
|
Corn |
63.7 |
68.9 |
Response: Thank you so much for pinpointing this valid comment. We have revised it accordingly.
Tables 2-8. Based on the standard errors provided, all values not marked with letter indexes are statistically insignificant. They must be marked with the same letter index. For example (Table 2, first row):
|
Initial |
54.88a |
54.88a |
54.89a |
Response: Apologize. Usually, we put a,b markers only for statistical significant values. We realize the importance of markers from your valid comment. As per your comment we have marked (table 2-8) statistically insignificant values with same indexes.